# Performance Testing and Rapid Solidification Behavior of Stainless Steel Powders Prepared by Gas Atomization

**DOI:** 10.3390/ma14185188

**Published:** 2021-09-09

**Authors:** Hang Qi, Xianglin Zhou, Jinghao Li, Yunfei Hu, Lianghui Xu

**Affiliations:** 1State Key Laboratory for Advanced Metals and Materials, University of Science and Technology Beijing, Beijing 100083, China; S20181191@xs.ustb.edu.cn (H.Q.); S20171206@xs.ustb.edu.cn (Y.H.); S20171236@xs.ustb.edu.cn (L.X.); 2Department of Mechanical Engineering, McGill University, Montreal, QC H2A 0C3, Canada; jinghao.li@mail.mcgill.ca; 3Chengdu Holy Industry Co., Ltd., Chengdu 610000, China

**Keywords:** gas atomization, rapid solidification, microstructure

## Abstract

Gas atomization is a widely used method to produce the raw powder materials for additive manufacturing (AM) usage. After the metal alloy is melted to fusion, gas atomization involves two relatively independent processes: liquid breakup and droplet solidification. In this paper, the solidification behavior of powder during solidification is analyzed by testing the powder’s properties and observing microstructure of a martensitic stainless steel (FeCrNiBSiNb). The powder prepared by gas atomization has high sphericity and smooth surface, and the yield of qualified fine powder is 35%. The powder has typical rapid solidification structure. Collision between powders not only promotes nucleation, but also produces more satellite powder. The segregation of elements in powder is smaller as the result of high cooling rate which can reaches 4.2 × 10^5^ K/s in average. Overall, the powder prepared by gas atomization is found to have good comprehensive properties, desired microstructure, and accurate chemical component, and it is suitable for various additive manufacturing techniques.

## 1. Introduction

Metal powders are increasingly used in powder metallurgy [1], 3D printing [2], laser cladding [3], and powder injection molding [4]. In order to obtain better products, it is necessary to use powders with better properties, including high sphericity, appropriate particle size distribution, and low impurity element contents. Gas atomization is currently one of the most widely used methods to prepare metal powders. These metal powders have good sphericity, small particle sizes, concentrated particle size distributions, and can be conveniently controlled by adjusting process parameters. Although the gas atomization method has been studied for more than 90 years [5], the breakup mechanism during atomization is still not clear. The fast fluid velocity, high temperature, and the lack of an effective observation method make it complex and difficult to research. It is generally belbived that the gas atomization process involves four stages [6,7]: (1) the break-up of a liquid column into ligaments; (2) the break-up of ligaments into droplets, which is known as primary breakup; (3) the breakup of droplets into smaller droplets, which is called secondary breakup; (4) the spheroidization and solidification of the droplets into powders.

The process of gas atomization can generally be divided into two processes: atomization breakup and droplets solidification. Atomization breakup is often studied by computer numerical simulation. Many studies have used computational fluid dynamics (CFD) software [8], like Fluent of ANSYS (ANSYS, Inc., Canonsburg, PA, USA), to calculate the influence of various parameters on atomization flow fields [9,10,11]. There are also some analysis methods based on optical principles for measuring the shape and velocity of particles in a flow field, which is called visualization research, e.g., high-speed imaging, particle imaging velocity field meters, etc. [12,13,14].

The droplets will solidify after completely broken. Due to the small particle size and fast heat transfer rate, the cooling rate can usually reach 10^5^ K/s. At present, it is generally calculated by numerical simulation, as the result of the difficulty to determin the cooling rate during the atomization process. When the cooling rate of a droplet is fast enough, it exceeds the critical rate of rapid solidification [15,16], and the resulting microstructure displays unique characteristics. However, the smaller particle size of the powder not only increases the cooling rate, but also decreases the nucleation probability. And, the nucleation probability decreases with the decrease of the diameter. After a large number of atomization experiments, Libera et al. [17] proposed that the solidification during an actual atomization process is mainly heterogeneous nucleation. Shi et al. [18] studied the influence of different parameters on powder shape, such as atomization gas, metal composition, nozzle structure. Pan et al. [19] studied the effect of particle size on the microstructure of the powder. Few studies [20] have investigated the nucleation mechanism during droplet solidification. Studying the nucleation mechanism during rapid solidification of powders helps to understand the reasons why droplets have high solidification rates. It is necessary to research the nucleation of powder because nucleation is related to the rapid solidification process. The existence of satellite powder will seriously affect the fluidity and other properties of powder. How to reduce the number of satellite powder is a concern of researchers. The rapid solidification of powders results in a unique metallographic structure, and the rapid cooling rate also influences the distribution of elements in the powders. Studying the microstructure and segregation of powders is helpful to better study the rapid solidification process.

## 2. Materials and Methods

This study is based on an iron-based multi-component alloy powder (FeCrNiBSiNb) for laser cladding. This material is newly developed and often used as a coating material for workpieces with high loads such as hydraulic supports, and its service state is martensite. The particle size, spherical degree and composition of powder have a great influence on the subsequent use. The particle size distribution of the powder is determined by laser particle size and particle shape analyzer (Bettersize 3000 plus, Bettersize Instruments Ltd., Dandong, China). And the surface morphology and sphericity of the powder are observed by SEM (SUPRA55, ZEISS, Stuttgart, Germany), and the elemental content of the powder is determined by atomic absorption spectrometer and oxygen-nitrogen-hydrogen analyzer. Through the test of these properties, the quality of the powder can be reflected better. In addition, the phase, microstructure, and element distribution of the powder were analyzed to study the solidification process of the droplet. The phase of the powder is detected by XRD (SmartLab, Rigaku, Japan). The content of alloying elements, oxygen elements, and hydrogen elements is measured by atomic absorption spectrometer (MKIIM6, THERMO, Waltham, MA, USA) and oxygen-nitrogen-hydrogen analyzer (ONH-2000, ELTRA, Retsch-Allee, Germany). Metallographic samples were prepared from the crosssection of the powder and corroded with ferric chloride-hydrochloric acid. The metallographic structure of the powder was observed and the element distribution of the powder was analyzed by EPMA (JXA-8100, JEOL, Tokyo, Japan).

## 3. Results

### 3.1. Size Distribution of the Powder

The powder obtained by atomization is shown in the Figure 1. The particle size distribution of the powder is wide, and the powder size is between 0 and 800 μm. There are large size powder particles and unbroken sheet metal which can be distinguished by naked eyes. The existence of sheet metal also proves that there is a process in which the liquid flow decomposes into film and then breaks into strips.

The particle size distribution of the original powder produced by gas atomization generally follows the normal distribution, and the range of particle size distribution is wide. The powder that meets the particle size requirements can only be obtained after multi-stage screening. The particle size distribution of powder was detected by laser particle size and a particle shape analyzer, and the results are shown in the Figure 2. The mean (average) particle size *d*_50_ is 159 μm with a standard deviation of 1.63. According to the application requirements of laser cladding, the target powder size should be between 53 μm and 149 μm, and the powder yield is about 35%.

### 3.2. Sphericity and Surface Morphology of the Powder

The surface morphology of the powder was observed by SEM, as shown in Figure 3 and Figure 4. The powder has good sphericity, smooth surface and most of the powder is close to spherical shape. At the same time, there are some non-circular powders and satellite powders, which become less round with the increase of particle size. And the powder is almost round when the particle size is small enough. Before the powder has not completely solidified, the droplets gradually become spherical due to the surface tension. If droplets solidify before spheroidization, they retain their previous shape and form ellipsoidal or other powders. By comparing powders of different sizes, as shown in the Figure 3, the smaller the particle size, the higher the sphericity of the powders, the less likely the abnormal or satellite powders are to occur. This is due to the shorter cooling and spheroidizing time of the powder with small particle size. After spheroidizing cooling, the impact of other particles on the morphology of the powder is relatively small. However, the cooling time and spheroidizing time of larger powder particles are more, and the collision of other particles will greatly affect the surface morphology of the powder when it is not fully solidified. At the same time, with the increase of particle size of the powder, the probability of collision of the powder increases greatly. Satellite powder is almost unavoidable for powder particles with a certain size. According to a large number of observations and statistics, this critical diameter is about 100 μm.

### 3.3. Phase Analysis of the Powder

The powder is divided into three groups according to grain size, powder A (*d*_50_ = 11.81 μm), powder B (*d*_50_ = 103.3 μm) and powder C (*d*_50_ = 337.4 μm). Group B is the target powder of the product, group A and group C are the powder with particle size less than and greater than the target powder, respectively. And a group of as-cast alloy samples with the same composition are added. The XRD was used to test the samples separately and the material of the tube anode is Cu. The peak image is shown in the Figure 5.

There is only a body-centered cubic structure in the powder A. According to the composition of stainless steel, the structure is martensite and ferrite. In the powder B and powder C, in addition to body-centered cubic structure, there is also face-centered cubic structure, which is austenite. The peaks of two different crystal structures deviate from the spectral lines of standard PDF cards to some extent due to the solubility of some Cr and Ni atoms in the lattice, which results in the change of lattice parameter. In as-cast samples, both body-centered cubic and face-centered cubic structures are ferrite and austenite, with the lower peak being carbide (Fe,Cr)_7_C_3_. Austenite occurs in powders with a larger particle size due to the following three reasons: (1) The cooling process of droplets with larger particle size is slower than that of droplets with smaller particle size; the austenite phase grows to a larger size; the transformation of austenite is incomplete during subsequent martensitic transformation and some austenite is retained; (2) Due to the use of nitrogen as atomizing gas, element N diffuses into the droplets at high temperature and is transferred from N element, it is stable austenite element, austenite zone expands, resulting in partial austenite appearing in normal temperature structure; (3) Austenite forming elements accumulate in the dendritic core, causing austenite content to increase.

### 3.4. Composition and Oxygen Content of the Powder

During metal smelting and solidification, there may be loss of its own elements and introduction of impurities. Changes in the content of elements and the presence of impurities may cause the performance of powders to be inconsistent with the design expectations. The content of alloying elements, oxygen elements, and hydrogen elements is measured by atomic absorption spectrometer and oxygen-nitrogen-hydrogen analyzer. The analysis results are shown in the Table 1.

It can be seen from the data in the table that impurities of O and N elements are introduced into the powder during the milling process. In addition, some losses of Si and B elements result in reduced content and other alloying elements contents is slightly higher than the design content. Among them, O element may be introduced in the atomization process and subsequent storage and transportation process. With the increase of powder size, oxygen content has an upward trend. Because of the larger specific surface area of the powder with small particle size, it is easier to oxidize during atomization and subsequent storage, but the overall oxygen content of the powder is low, which has little influence on subsequent use. Due to the use of nitrogen as protective atmosphere and atomizing gas, N element is easy to dissolve in metal melt and may be solid soluble in matrix, as gap atom after solidification. In addition, N element is easily introduced during screening, storage, and use as impurity elements. A certain amount of N element can improve the mechanical properties of stainless steel [21]. Elements Si and B may react to form other compounds during the smelting process, resulting in low element content in the final powder. The content of other elements is not significantly different from that of the design and is close to that of powders with different grain sizes.

### 3.5. The Microstructure of the Powder

Many results of computer numerical simulations and experiments showed that the diameter of the broken droplets is very small, and the flight speed is very fast. The solidification speed reached 10^5^–10^6^ K/s during the solidification process [22], so the atomization is a typical rapid solidification process. The extremely fast cooling rate makes the microstructure of the solidified powder different from that obtained at a slow cooling rate.

The metallographic structure of the sample was observed by a scanning electron microscope. In the Figure 6, a powder particle with a diameter of about 280 μm is magnified by 1000 times. The particle contains a large number of secondary dendrites, as well as some primary dendrites and many fine cellular crystals. The secondary dendrite has no obvious orientation, so nucleation occurred in several places at the same time when droplets solidified. According to the phase diagram and solidification process analysis, the microstructure in the grains was martensite, ferrite, and retained austenite, and the interdendritic structure was mainly austenite and precipitated phases.

The microstructure of the powder is compared with that of an ingot with the same composition, as shown in the Figure 7. Although there are secondary dendrites in both powders and ingots, their size in the ingots is much larger than that in powders. Due to the slow cooling rate, some pearlite appeared between secondary dendrites in the ingot, which was not observed in the powders; therefore, changing the cooling rate led to different solidification process.

Analysis of a large number of powder microstructure photos showed many cases of collision and combination between large and small particles. There are three kinds of combination forms in Figure 8: not embedded, partially embedded, and completely embedded.

In Figure 8a, the small particle powder is completely outside the large particle powder, and there is a transition zone between the small and large particle powders, whose grain size is closer to the large particle powder; therefore, collisions may occur when the small particles are fully solidified, but when the large particles are not yet fully solidified. Under the action of surface tension, the non-solidified liquid phase coats the small particles and forms a neck-like structure. Due to the late collision time, the small particles could not participate in the solidification process of the large particles as external nucleation particles, which means that the solidification of the large particles was not affected. Metallurgical bonding also formed between the large and small powders, which was not broken during subsequent screening and other processes. This kind of combination form displays a typical satellite powder structure, which negatively impacts the powder’s fluidity. There are also some partially-embedded powders in the Figure 8b, which have an obvious interface but no transition zone. This is similar to the completely embedded powder, but with an earlier collision time during solidification. As shown in Figure 8c, there is a completely embedded powder with a diameter of about 16 μm near the surface. During the solidification of the large particle, the small-particle, as an exotic nucleation particle, grows a large number of secondary dendritic structures all around. Secondary dendrites grown from small-particle displayed an obvious orientation. Secondary dendrites near the small-particle powder began to grow after the powders were embedded, and the large droplets did not solidify completely. Collisions may occur in the early stage of droplet solidification when there is more liquid phase. This may also explain why small-particle powders were completely embedded in the droplets.

According to the analysis of the above powder microstructures, the combination of large and small powders after collision was mainly related to the time of collision. In general, when collisions occur, small-particle powders are almost completely solidified due to their faster cooling rate. If large-particle contains more liquid phase during collision, the small-particle is more easily embedded in the droplet and participates in the solidification process as an exotic nucleation particle. If there is little liquid phase and the small-particle is partially embedded or stays on the surface during collision, it becomes partially embedded or not embedded, which has limited influence on the solidification of the remaining liquid phase. The contact between the small-particle and liquid phase can help nucleate the liquid phase and accelerate the solidification process. This influence is related to the amount of remaining liquid phase.

### 3.6. Nucleation Characteristics of the Droplet

Droplet solidification can be divided into nucleation and growth processes. Non-uniform nucleation is more likely to occur due to the smaller critical nucleation radius. In general, inhomogeneous nucleation requires vessel wall or impurities to act as exogenous nucleation particles. The influence of vessel wall can be neglected during gas atomization. Therefore, inhomogeneous nucleation is mainly participated by solidified powders and impurities of droplets themselves as exogenous nucleation particles. It is generally considered that there are two ways to nucleate droplets [23]: (1) droplets with a diameter less than a certain size nucleate from the point on the surface; (2) droplets with a diameter greater than a certain size have many nucleation particles on the surface and grow inwards at the same time, as shown in the Figure 9.

A different nucleation mode was observed. As shown in Figure 10, the secondary dendrites in the powder show orientation and grow outwards from an origin. The solidified small-particle powder collides with the non-solidified droplet and enters into the droplet. The whole surface of the powder can be equated to the vessel wall participating in non-uniform nucleation. The subsequent grain growth direction is perpendicular to the surface of small particles, similar to that of dendrites growing perpendicular to the vessel wall during general solidification process. Secondary dendrite appears in the structure and the secondary dendrite arm space is small, which indicates that the cooling rate is faster. This nucleation mechanism differs from the above two in that its nucleation begins at the particles inside the droplet rather than the surface of the droplet. The heat transfer rate decreases with increasing droplet size. However, the higher probability of collision with small-particle powder, more the small-particle powder helps the droplet nucleate and ensures the cooling rate of the droplet. It is still possible to achieve rapid solidification.

Due to deep supercooling, droplets with very small particle size can complete homogeneous nucleation by component fluctuation and energy fluctuation; larger droplets mainly rely on their own impurities and small particle size powder as foreign nucleation particles to complete non-homogeneous nucleation. These nucleation methods ensure sufficient nucleation rate during droplet solidification, so that the cooling rate of droplets is higher than the critical cooling rate for rapid solidification, in order to obtain the rapid solidification structure at room temperature. Therefore, in the process of gas atomization, the nucleation of droplets is mainly non-uniform nucleation with a few droplets which may be uniform nucleation due to deep supercooling.

### 3.7. Elemental Distribution of Powders

During solidification, the redistribution of solute elements makes the composition of the newly-formed solid phase different from that of the surrounding liquid phase, resulting in an inhomogeneous chemical composition, which is called segregation. Segregation is often divided into macroscopic segregation and microscopic segregation, and it can lead to different properties of a material, including material failure. For example, a low chromium content at the grain boundary of stainless steel readily causes intergranular corrosion. Rapid solidification can prevent segregation to a certain extent due to its fast cooling rate and short diffusion time. The elemental distribution of the powder was analyzed by EPMA.

To compare differences in the elemental distribution of the powder and common solidification metal, a stainless steel ingot with the same composition was cast. A powder with a diameter of 19 μm, a secondary dendrite in a powder with a diameter of 104 μm, and a secondary dendrite in an ingot were selected to analyze the elemental distribution. The results are shown in Figure 11.

In the powder with a diameter of 19 μm, every element is well-distributed, mainly because the cooling rate is very fast, and the powder solidified before solute diffusion. According to the empirical formula [22], the average cooling rate is about 4.2 × 10^5^ K/s. Here is the empirical formula:(1)$$\dot{T}={\left(\frac{40}{0.00756d+0.3908}\right)}^{3},$$

The content of Cr in the interdendritic region is higher than in the dendrite, while the contents of Ni and Si in the interdendritic region are lower than in the dendrite. Si is enriched at some points in the ingot with the slowest average cooling rate of 2 K/s. To represent the degree of segregation of alloy elements, the ratio of the content of alloying elements in the core of dendrites, *C*_S_, to that in the interdendritic region, *C*_L_, is defined as the segregation coefficient *k*. The expression of *k* is:(2)$$k=\frac{C_{S}}{C_{L}}$$

When *k* < 1, the solute elements are positively segregated, which indicates that the elements tend to gather in the interdendritic region, and the smaller the *k* value, the greater the degree of segregation; when *k* > 1, the solute elements are negatively segregated, which indicates that the elements tend to gather in the dendrites. The higher the *k* value, the greater the degree of segregation; and if *k* = 1, there is no segregation.

The mass fraction of several easily-segregated elements in the dendrite core and interdendritic region was obtained, and the segregation coefficient of the elements was calculated. The results are shown in Table 2 and Table 3. Due to the faster solidification rate and shorter solute diffusion time of each element, the segregation of the powder was obviously improved compared with that of the ingot.

## 4. Discussion

In this study, the particle size distribution, morphology and phases of the FeCrNiBSiNb stainless steel powder were investigated. From a production point of view, the yield of qualified fine powder (15–53 µm) produced by gas atomization can reach 35%. The powder has high sphericity and smooth surface, and it meets the requirements of most subsequent processing and forming methods. The phase composition of the powder differs from that of the as-cast condition as a result of the rapid solidification and cooling effect. It will have an influence on the subsequent forming methods in which powder is incomplete melt during process.

The metallographic structure of the powder was observed, and a typical rapid solidification structure (dendritic/cellular structure) was found. Microstructure is mainly composed of fine secondary dendrites and fine cellular crystals. In the powders with small diameter, there are only cellular crystals. Collisions between powders during atomization are very common. On the one hand, the solidified small powders provide additional nucleation particles for the non-solidified droplets by collision, which increase the cooling rate. On the other hand, excessive collisions also increase the formation probability of satellite powder, which seriously degrades the fluidity of the powder. Increasing the cooling rate of droplets by technology improvement may decrease the collisions at the end of solidification that decrease the formation of satellite powders. Further research is necessary in the future.

The segregation of each element was analyzed. By testing the distribution of powder elements, it is found that different elements have different segregation tendencies. Cr element is positive segregation element, it tends to concentrate between dendrites; Ni element and Si element are negative segregation element and tend to concentrate on dendrite branch. By calculating the segregation coefficient of elements, the segregation of elements in powder is obviously smaller than that of as-cast alloy. Smaller-sized powders have uniform distribution of alloying elements without obvious segregation, and this may attribute to the fast cooling rate that inhibits solute diffusion. In addition, powder with small segregation is helpful to the cladding process of powder incomplete melting on the performance, such as cold spraying.

## 5. Conclusions

Gas atomization is a good method to prepare metal powder. The powder has high sphericity and smooth surface. And in this research, the yield of qualified fine powder is 35%.The powder has typical rapid solidification structure. Collision between powders is one of the important mechanisms for droplet nucleation and the formation of satellite powders.The segregation of elements in powder is smaller than that of as-cast alloy. The main reason is that the cooling rate of the powder is relatively high. In this research, the average cooling rate of the powder can reaches 4.2 × 10^5^ K/s.

## Figures and Tables

**Figure 1 materials-14-05188-f001:**
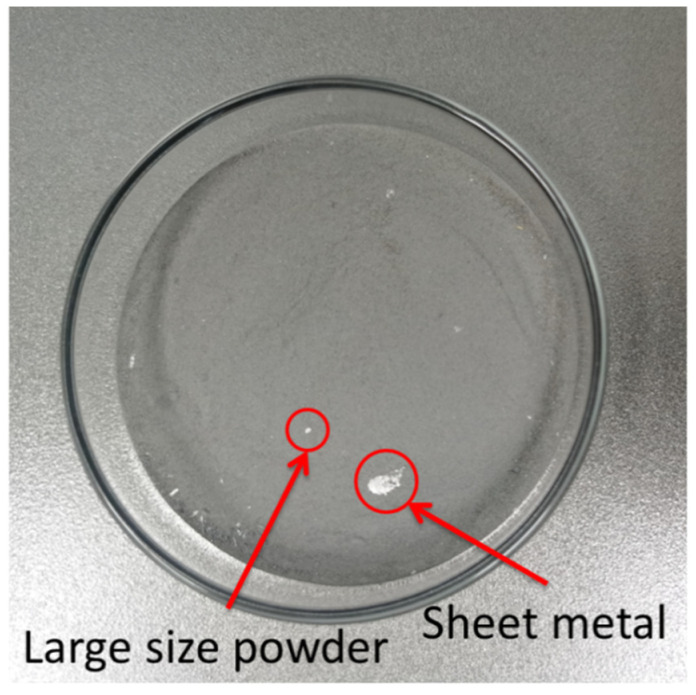
Original powder.

**Figure 2 materials-14-05188-f002:**
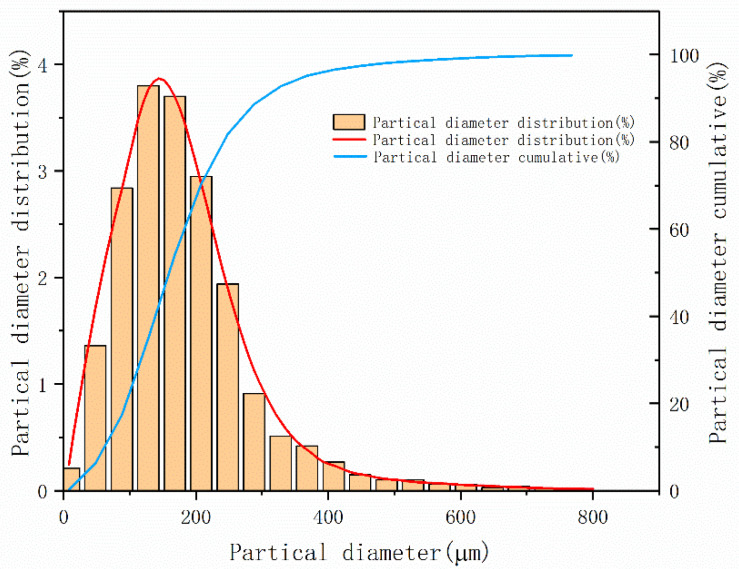
Particle size distribution of original powder.

**Figure 3 materials-14-05188-f003:**
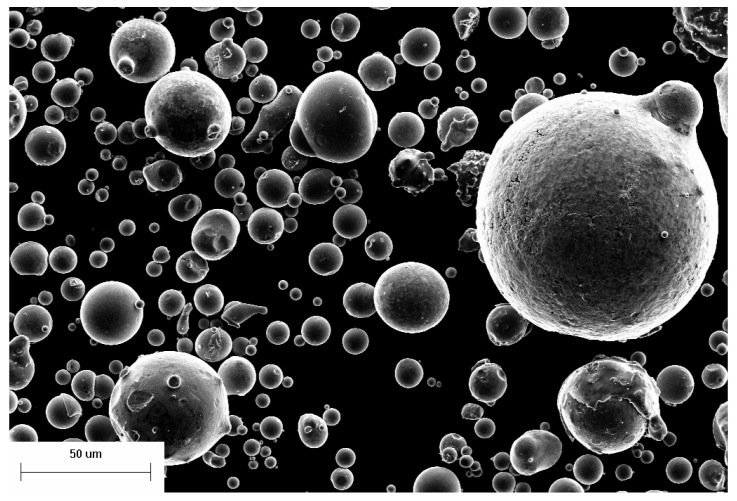
Morphology of original powder.

**Figure 4 materials-14-05188-f004:**
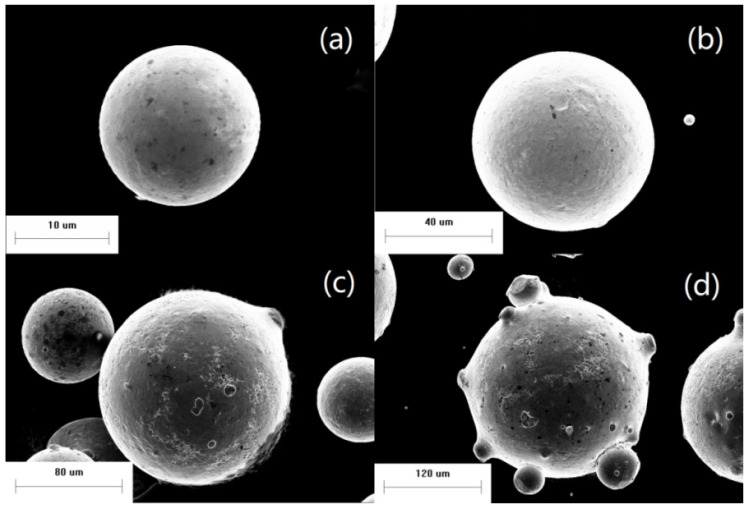
Single-particle morphology of powder: (**a**) *d* = 17 μm; (**b**) *d* = 69 μm; (**c**) *d* = 144 μm; (**d**) *d* = 229 μm.

**Figure 5 materials-14-05188-f005:**
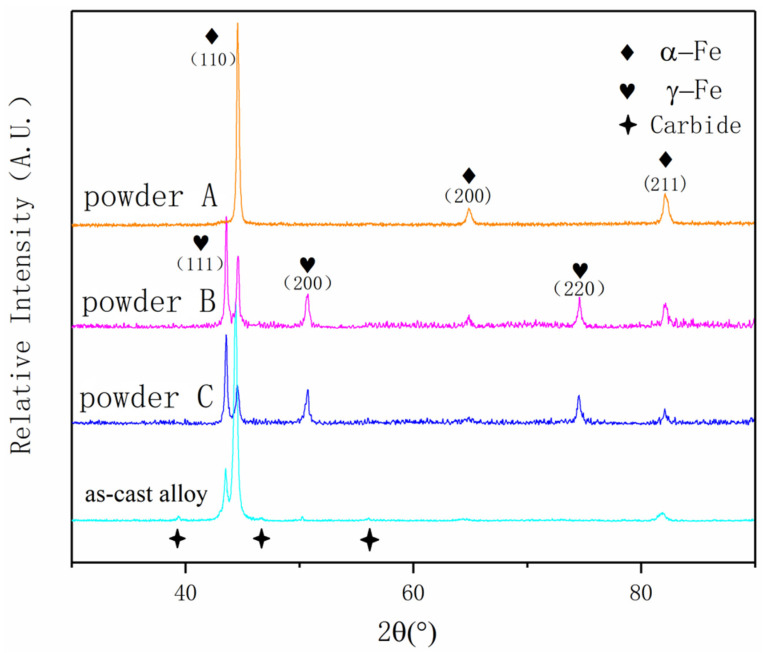
X-ray diffraction results of powders and as-cast alloy.

**Figure 6 materials-14-05188-f006:**
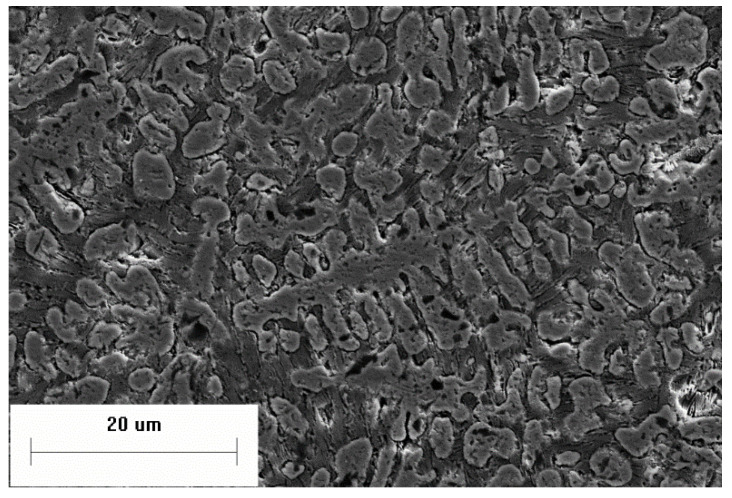
Powder microstructure (*d* = 280 μm).

**Figure 7 materials-14-05188-f007:**
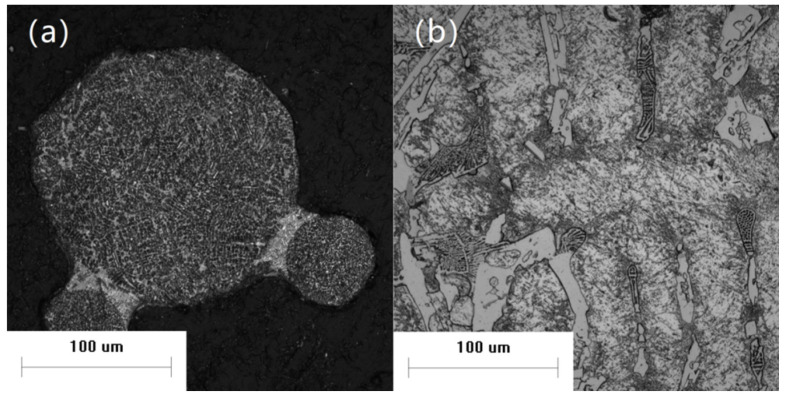
Microstructure of the (**a**) powder compared with that of the (**b**) ingot.

**Figure 8 materials-14-05188-f008:**
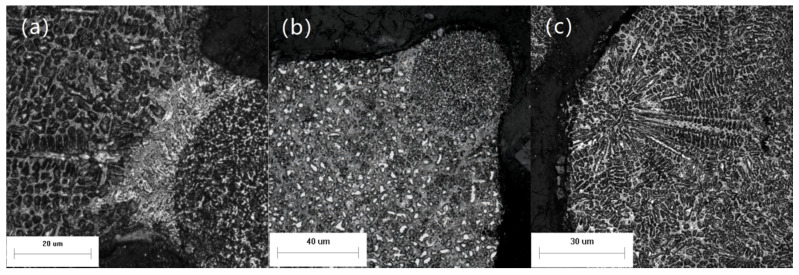
Typical combination forms of powders: (**a**) not embedded; (**b**) partially embedded; (**c**) completely embedded.

**Figure 9 materials-14-05188-f009:**
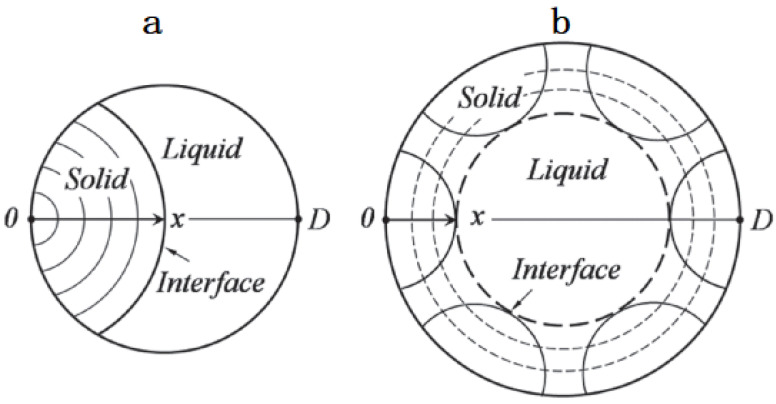
Schematic diagram of nucleation in liquid droplet: (**a**) single nucleation; (**b**) multiple nucleation.

**Figure 10 materials-14-05188-f010:**
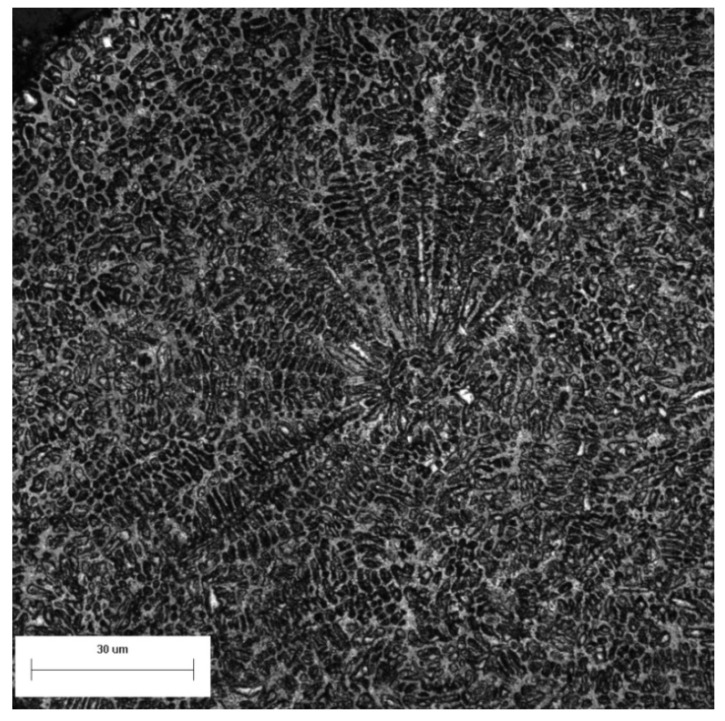
Microstructure of the powder.

**Figure 11 materials-14-05188-f011:**
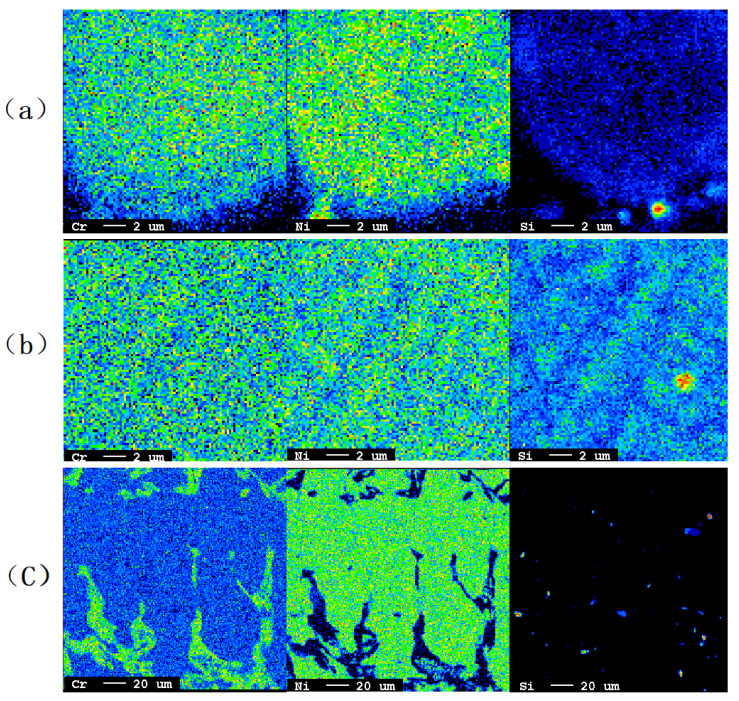
Distribution of elements: (**a**) powder with a diameter of 19 μm; (**b**) secondary dendrite in a powder with a diameter of 104 μm; (**c**) secondary dendrite in an ingot.

**Table 1 materials-14-05188-t001:** Element content of powder.

	O	N	Cr	Ni	Nb	Si	B
Design content	0	0	16%	4%	1%	1%	1%
*d*_50_ < 53 μm	0.11%	0.10%	17.1%	4.28%	1.02%	0.75%	0.69%
*d*_50_ = 53 μm–149 μm	0.026%	0.13%	16.8%	4.43%	1.22%	0.77%	0.69%
*d*_50_ > 149 μm	0.019%	0.13%	17.1%	4.39%	1.15%	0.76%	0.70%

**Table 2 materials-14-05188-t002:** Elemental contents of the powder and their segregation coefficients.

Microstructure	Fe	Cr	Ni	Si
Dendrite	75.97%	15.35%	4.59%	1.09%
Interdendritic region	76.40%	16.43%	3.32%	0.85%
Segregation coefficient	-	0.93	1.38	1.28

**Table 3 materials-14-05188-t003:** Elemental contents of the ingot and their segregation coefficients.

Microstructure	Fe	Cr	Ni	Si
Dendrite	76.32%	14.15%	5.46%	1.19%
Interdendritic region	73.41%	20.38%	1.90%	0.59%
Segregation coefficient	-	0.69	2.87	2.02

## Data Availability

Data sharing not applicable.

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
