# Peer review of "Performance Testing and Rapid Solidification Behavior of Stainless Steel Powders Prepared by Gas Atomization"

_materials, 2021, doi:10.3390/ma14185188_

Round 1

Reviewer 1 Report

The manuscript entitled: “Performance Testing and Rapid Solidification Behavior of Stainless Steel Powders Prepared by Gas Atomization” may be relevant for the Materials journal. The article based on original experimental research. The authors introduced a lot of changes compared to the previous version but some errors still occur.

  • Abstract – Please investigate and change the form of abstract. It looks like it's an introduction. Write more or less in this form - what was tested, how or what was tested, what came out - the most important results and some conclusion to it. The Abstract should be an abridged version of the article that encourages reading.
  • Line 42 – Please provide additional information about the production site of the used software
  • Line 48 – Please add a space between value and unit.
  • Line 58 – Removing etc. and just put a full stop.
  • Fragment 61-65 – It looks like a concussion. Please change it. Based on the introduction presented, please write what is the aim of your research.
  • Line 73 – Please provide additional information about the production site of the used measuring equipment - microscope. It is in the next part of manuscript but it should be here.
  • Line 78 – Please provide additional information about the production site of the used measuring equipment – x-ray diffractometer. It is in the next part of manuscript but it should be here.
  • Line 124 – Please provide additional information about the production site of the used measuring equipment – EMPA. It is in the next part of manuscript but it should be here.
  • Line 96 – Is this an analyzer model? Shouldn't there be any spaces?
  • Line 157 - Please provide additional information about the production site of the used measuring equipment – MKIIM6 and ONH-200. Why is there nothing about it in the materials and methods section. Please add there information about the equipment used.
  • Table 1 – No markings elements for which content are presented. Take a look at Table 3. There, the headings show the elements Fe, Cr, Ni, Si and their values. However, here we do not know what these values are for Design content, etc.
  • Line 321 – Lack of spacing between the text and the table.
  • Table 2 – No markings elements for which content are presented. Take a look at Table 3.
  • Chapter 4 should be called Conclusion. Move the discussion to chapter 3 and name this chapter "Results and discusion". Returning to the chapter Conclusion, it should contain the most important conclusions from the obtained research and, if possible, some specific numerical values that confirm it.

The article was prepared better than before. Please re-track the Materials template available on the website:

https://www.mdpi.com/journal/materials/instructions

There it describes what should be found in each section, and shows what the article should look like (abstract, introduction, “materials and methods”, “results and disusion” and conclusion). The presented conclusions are not very much supported by the results presented in the article. However, I am inclined to accept the article after making all the above-mentioned changes changes.

Author Response

Dear Reviewers:

Thank you for your letter and for the reviewers' comments concerning our manuscript entitled “Study on the Breakup and Solidification during Gas Atomization using High-speed Imaging and Microstructure Observation”(materials-1233805). Those comments are all valuable and very helpful for revising and improving our paper, as well as the important guiding significance to our researches. We have studied comments carefully and have made correction which we hope meet with approval. The main corrections in the paper and the responds  are as flowing:

The abstract has been revised and maybe  look more concise now. All writing mistakes have been corrected. The additional information about the production site of the used measuring equipment is in the right place now. According to Materials template, the discussion and conclusion are separated. I added a chapter of conclusion, the structure of the article should be more complete. In addition, there are some minor amendments not mentioned。 These changes will not influence the content and framework of the paper. Special thanks to you for your comments.

I appreciate for your warm work earnestly, and hope that the correction will meet with approval.

Once again, thank you very much for your comments and suggestions.

Yours sincerely,

Hang Qi

Reviewer 2 Report

powder automisation is an important topic

literature review is good

materials and methods are bad, need revision

results are good

discussion is strange, usually discussion is about comparison with the previously published data, or some structure formation mechanisms, which will be difficult to discuss here because of high and unknown cooling rate, in this paper some computer modeling of the phase balance might improve the presentation.....at the moment it looks like a summary of results

conclusions are not presented

on a whole, this manuscript looks like a good first draft

please see the pdf file for detailed comments

Author Response

Dear Reviewers:

Thank you for your letter and for the reviewers' comments concerning our manuscript entitled “Study on the Breakup and Solidification during Gas Atomization using High-speed Imaging and Microstructure Observation”(materials-1233805). Those comments are all valuable and very helpful for revising and improving our paper, as well as the important guiding significance to our researches. We have studied comments carefully and have made correction which we hope meet with approval. The main corrections in the paper and the responds  are as flowing:

The part of materials and methods has been revision. It is now more detailed and reasonable.  I added a chapter of conclusion, the structure of the article should be more complete. Limited by some conditions, I may not be able to add additional research  to the paper. In addition, there are some minor amendments not mentioned. These changes will not influence the content and framework of the paper. Special thanks to you for your comments.

I appreciate for your warm work earnestly, and hope that the correction will meet with approval.

Once again, thank you very much for your comments and suggestions.

Yours sincerely,

Hang Qi

Reviewer 3 Report

Probably, the work can be useful to researchers developing technologies for the production of powder steels for additive technologies. Can be recommended for publication.

A few technical notes:

  1. Line 66. It is not clear whether the investigated powder was produced by the authors of the work or some kind of industrial material was taken. In the first case, apparently, technical conditions should be specified in as much detail as possible, since they, obviously, can affect the investigated characteristics of the resulting powder (after all, this is a scientific work, not an advertisement for know how). In the second case, the question arises why all of this (or part of it) has not been studied earlier, since this material is already being produced by someone.
  2. Line 97 “The particle size d50 is 159 μm” → “The mean (average) particle size d50 is 159 μm”?
  3. Line 104 “The powder has better sphericity” Better than what?
  4. Line 107 “and are almost round when the particle size is smaller.” Smaller than what?
  5. Line 128. It is not clear on what basis these three groups were distinguished.
  6. Line 130 or fig. 5. It would also be nice to indicate the wavelength or spectrum line and the material of the tube anode
  7. Lines 160 and 321. Names of elements disappeared in the table.

Author Response

Dear Reviewers:

Thank you for your letter and for the reviewers' comments concerning our manuscript entitled “Study on the Breakup and Solidification during Gas Atomization using High-speed Imaging and Microstructure Observation”(materials-1233805). Those comments are all valuable and very helpful for revising and improving our paper, as well as the important guiding significance to our researches. We have studied comments carefully and have made correction which we hope meet with approval. The main corrections in the paper and the responds are as flowing:

The materials is newly developed by a company I work with. For reasons of confidentiality, I can't mention the name of the company and the specific composition of the materials. Please forgive me. Some inappropriate places and wrong contents have been revision. The basis of grouping has been added to the paper. I'm sorry for my negligence. All other  mistakes have been corrected. In addition, there are some minor amendments not mentioned. These changes will not influence the content and framework of the paper. Special thanks to you for your comments.

I appreciate for your warm work earnestly, and hope that the correction will meet with approval.

Once again, thank you very much for your comments and suggestions.

Yours sincerely,

Hang Qi

Round 2

Reviewer 2 Report

could be better

This manuscript is a resubmission of an earlier submission. The following is a list of the peer review reports and author responses from that submission.

Round 1

Reviewer 1 Report

The manuscript entitled: “Study on Performance Testing and Solidification Behavior of Stainless Steel Powders Prepared by Gas Atomization” may be relevant for the Materials journal. The article based on original experimental research. However, the article required a lot of improvements:

  • Line 24, 25 etc. – no literature references. Please write references in plain text and not as a reference in the text editor. It may also be related to incorrect formatting of references in the References section.
  • Line 34 – Please provide references from the use of this method, since it has been used for 90 years.
  • Line 44 – Please mention the manufacturer of the Fluent software if used.
  • Line 50 – extra dot.
  • Line 50 – “...cooling rate is very high...” - but how big is it? Please provide some value range.
  • Line 58 – “...and so on.” Please use a more formal phrase. In this respect, trace the entire manuscript.
  • Line 59 – "...few studies" - Please provide references.
  • Line 68 – The chapter should be named material and methods. Incorrect content organization.
  • general note - References in the text to figures in the Materials journal are written as Figure not Fig. - I quote: All figures and tables should be cited in the main text as Figure 1, Table 1, etc. Please write in accordance with the journal template.
  • general note - Please use the spaces between the text and drawings, and if they are cantered relative to the column, please remove indentation - the first line, because the drawings seem shifted to the right.
  • Line 79 – “...wide range of particle size...” – Please provide numerical values.
  • Line 88 – Please provide additional information about the production site of the used measuring equipment - microscope.
  • general note – according to the template - drawing captions should be justified between the left and right margins of the page.
  • Line 112 – different values for d50. Is not that a mistake? Why were so named. May choose a different name or other markings related to grain size. In its current form, is misleading.
  • Line 114 – Please provide additional information about the production site of the used measuring equipment – x-ray diffractometer.
  • Line 124 – We don't write lattice constant, but lattice parameter instead. This value is not constant and depends on many factors - temperature, pressure or alloy additions that cause changes in the crystal lattice. Therefore, the term "constant" cannot be used.
  • Line 140 – “...measured chemically” - what method was used, what apparatus was used. Nothing about it.
  • Line 141- Lack of spacing between the text and the table.
  • Table 1 – No markings elements for which content are presented.
  • Line 158 – the chapter should be called Results and discussion.
  • Line 173, 181, 187 – Lack of spacing between the text and the figures.
  • Line 229 – Please remove the phrase "…and work".
  • Line 240 – the description in the text refers to Figure 9, not Figure 10.
  • general note - according to the template, symbols (a), (b) etc. in the description of drawings should be written in bold.
  • Line 246 – no verb in the sentence - the fragment does not make sense.
  • Line 247 – Please write the higher probability - unnecessary “the” inserted before probability.
  • Line 251 – The figure shows the microstructure of powder. However, there is no reference to it in the text. Maybe the drawing is unnecessary.
  • Line 267 – "Diffusion time is very short" - how short is it? Was it measured to put such a conclusion.
  • EMPA – Please provide additional information about the production site of the used measuring equipment.
  • Line 283 – where are figures 12 and 13. Maybe it was figure 11 but you forgot to insert parts (a), (b) and (c). Move the drawing away from the right margin - center or align to the left without indenting the first line.
  • Line 287 i 290 – „cooling rate is very fast”, „slowest cooling rate” – lack of value, which makes it difficult to determine what is fast and what is slow.
  • Line 293 – Please correct the equation.
  • Line 299 – You should write dendrite core, not trunk. The introduced term is used less frequently.
  • Line 304 i 305 – Lack of spacing between the text and the table.
  • Line 304 – The lack of signs of elements in the table header.
  • Line 307 – what properties were measured? The article presents mainly the morphologies of the powder.
  • Line 316 – "There are a lot ...". The sentence should begin with a capital letter.
  • Line 326 – The term 'dendrite branch' is used, not stem.
  • Line 330 – different font.
  • References made contrary to the journal's guidelines, namely:

[1], [2], [4] - lack of volume and range of pages,

[5] – it's probably a book, so no publisher,

[9] – capital letters used in journal name,

[15] – it is a book or journal, maybe post-conference materials,

[21] – the lack of a range of pages.

The article was prepared very carelessly, contrary to the Materials template available on the website:

https://www.mdpi.com/journal/materials/instructions

One gets the impression that this is a fragment of a doctoral dissertation. This is indicated by the lack of the section: Materials and methods, and Results and discussion. The presented conclusions are not very much supported by the results presented in the article. Therefore, I am inclined to reject the article in its present form because it still requires a lot of work.

Reviewer 2 Report

The authors provide some analytical results of gas atomized powder of stainless steel. Unfortunately, the paper does not convey their results to the readers due to its poor description. The authors should check the paper carefully and should realize that some links for references are broken in the introduction before submission.  The names of elements were missing in Tables 1 and 2. Figures 12 and 13 were also missing, thus, I cannot determine whether the authors' mention is appropriate or not. Maybe, Fig. 10 is also inappropriate because it is impossible to understand which is single nucleation or multiple nucleation. 

The purpose of this study is not explained well either. The authors mention that the nucleation mechanism during droplet solidification should be investigated, but there is no explanation why and how. Is it possible to reduce the number of satellite particles by understanding the nucleation mechanism? Is it possible to enhance the powder flowability by clarifying the nucleation mechanism? If no, I cannot find any merits to clarify the metal structures and the elemental distributions because the powder will be molten again in cladding process. Only with the presented results, the authors cannot show the necessity of this study, i.e., investigation on the nucleation mechanism.

Including the insufficient preparation of submission, I cannot admit this paper as an academic report.